# Biomarkers of Acute Lung Injury The Individualized Approach: for Phenotyping, Risk Stratification and Treatment Surveillance

**DOI:** 10.3390/jcm8081163

**Published:** 2019-08-03

**Authors:** Daniel D. Murray, Theis Skovsgaard Itenov, Pradeesh Sivapalan, Josefin Viktoria Eklöf, Freja Stæhr Holm, Philipp Schuetz, Jens Ulrik Jensen

**Affiliations:** 1PERSIMUNE, Department of Infectious Diseases, Rigshospitalet, DK-2100 Copenhagen, Denmark; 2Respiratory Medicine Section, Department of Internal Medicine, Herlev-Gentofte Hospital, DK-2900 Hellerup, Denmark; 3Medical University Department, Kantonsspital Aarau, 5001 Aarau, Switzerland; 4Faculty of Medicine, University of Basel, 4001 Basel, Switzerland

**Keywords:** acute lung injury, biomarkers, omics

## Abstract

Do we need biomarkers of lung damage and infection: For what purpose and how should they be used properly? Biomarkers of lung damage can be used for diagnosis, risk stratification/prediction, treatment surveillance and adjustment of targeted therapy. Additionally, novel “omics” methods may offer a completely different and effective way of improving the understanding of pathogenesis of lung damage and a way to develop new candidate lung damage biomarkers. In the current review, we give an overview within the field of acute lung damage of (i) disease mechanism biomarkers, (ii) of “ready to use” evidence-based biomarker-guided lung infection management, (iii) of novel strategies of inflammatory phenotyping and how this can be used to tailor corticosteroid treatment, (iv) a future perspective of where “omics” technologies and mindsets may become increasingly important in developing new strategies for treatment and for understanding the development of acute lung damage.

## 1. Introduction; Acute Lung Injury and Biomarkers to Characterize This Condition, and to Assist in Treatment Strategies

Acute lung injury is characterized by hypoxemia, bilateral opacities on chest imaging like chest X-ray or CT-scans and the condition should not be fully explained by cardiac failure or fluid overload. The pathophysiology is not fully understood, but it often (but not always) includes an inflammatory process in the lungs, initiated by a number of etiologies like infection (pneumonia, sepsis), aspiration (gastric juices, water, etc.), trauma (lung contusion, major thoracic surgery), inhalation toxicity, blood transfusion and others. Pathologically, acute lung injury can be divided into an early phase (exudative) and a late phase (fibroproliferative), these include epithelial and endothelial damage, activation of inflammatory mediators and eventually, fibroblast activation, coagulation, cell proliferation and apoptosis pathways. 

In this review, we attempt to give the reader an insight into different biomarkers of acute lung injury. Since a comprehensive review of all biomarkers of lung injury would not fit into a single article, we focus on the following areas where biomarkers may be of special importance, and in which specific biomarkers have already been thoroughly investigated: (i) biomarkers of alveolar and bronchiolar damage, (ii) biomarkers of endothelial damage, (iii) biomarkers of lung infection, (iv) biomarkers to characterize phenotypic lung inflammation. Further, we present our view of some key points in how the ‘OMICS’-era can influence this important field. We realize, our choices regarding biomarkers described could have been different. Other promising biomarkers of lung injury do, in fact, exist; we have chosen some model biomarkers we think have a realistic clinical potential, either already established, or biomarkers that could come into clinical use within few years. On the other hand, we have put less weight on describing biomarkers that are less well established and validated, and biomarkers where a realistic assay for bioanalysis for daily practice has not yet been established, see Table 1. 

Biomarkers of lung injury can be used for diagnosis, risk stratification/prediction, treatment surveillance and adjustment of targeted therapy. Additionally, novel “omics” methods may offer a completely different and effective way of improving the understanding of pathogenesis of lung injury and a way to develop new candidate lung injury biomarkers. 

In the current review, we give an overview within the field of acute lung injury of (i) disease mechanism biomarkers, (ii) of “ready to use” evidence-based biomarker-guided lung infection management, (iii) of novel strategies of inflammatory phenotyping and how this can be used to tailor corticosteroid treatment, (iv) a future perspective of where “omics” technologies and mindsets may become increasingly important in developing new strategies for treatment and for understanding the development of acute lung injury. 

## 2. Biomarkers of Alveolar and Bronchiolar Injury—Surfactant Protein D, Club Cell Secretory Protein and Others

The barrier between air and the blood is constituted by three principal layers—the alveolar epithelium, the interstitial space including the basal membrane, and finally, the vasculature including the endothelium. Several lung diseases have, as a hallmark feature, the breakdown of this barrier. In the later years, specific biomarkers for lung injury have been identified with the intention to guide a more pathophysiological stratification of patients, in turn, allowing a personalized treatment for patients with lung injury. The most promising markers include surfactant protein D (SPD), club cell secretory protein 16 (CC16) and actin-scavenger gelsolin specific for alveolar, bronchial and endothelial damage to the lungs, respectively [1,2,3,4], Figure 1.

### 2.1. Biomarkers of Alveolar Injury

SPD is produced by type II pneumocytes and secreted to the surfactant layer of the alveoli. SPD enters the circulation when the alveolar air–blood barrier is injured and becomes detectable in plasma [2]. SPD can identify, and risk stratify patients with chronic obstructive pulmonary disease, pneumonia, idiopathic pulmonary fibrosis, critical-illness-associated respiratory failure, and correlate with severity of illness [5,6,7].

Krebs von den Lungen-6 (KL-6) has shown the same ability to indicate alveolar injury as SPD. KL-6 is a mucin-like glycoprotein produced by type II pneumocytes, and patients with acute respiratory distress syndrome (ARDS, the most advanced form of acute lung injury) have increased KL-6 in plasma compared to healthy controls [8]. 

Soluble receptor for advanced glycation end-products (s-RAGE) is primarily produced by type I pneumocytes in the alveoli, and an association between increasing plasma RAGE and alveolar injury has been examined recently. The clinical usefulness of this biomarker is not fully elucidated yet, but plasma RAGE and severity of lung injury have been shown to correlate [9]. 

Systemic inflammation is both a common cause and consequence of lung injury. Thus, unsurprisingly, several inflammatory markers have been associated with lung injury. Including alveolar injury in patients with idiopathic pulmonary fibrosis and the interleukins IL-6 and IL-8 [10], and mortality in patients with ARDS [11]. 

### 2.2. Biomarkers of Bronchiolar Injury

There are few biomarkers of injury involving the bronchi and bronchioles. Plasma concentration of club cell secretory protein 16 (CC16) increases due to increased permeability of the air–blood barrier within the conductive airways where the club (or clara) cells responsible for CC16 production are located [12]. Plasma CC16 can be used to evaluate severity of asthma exacerbations, as plasma CC16 increases in allergic asthmatics exposed to their allergen [13]. Although some studies have shown a diagnostic value of CC16 for diagnosis of ARDS [14], a large study did find that CC16 was not useful in predicting respiratory prognosis among critically ill mechanically ventilated patients [1]. CC16 blood levels have been demonstrated to be strongly associated to the degree of lung function reduction in patients with chronic obstructive lung disease [15], and thus, may have a larger potential in the non-acute setting. CC16 will not be described in further detail in this review. 

### 2.3. Biomarkers of Endothelial Injury

Actin-scavenger gelsolin is produced by muscle cells continuously and diffuses into circulation to serve as a physiological buffer for actin release [16,17]. As tissue damage leads to release of actin, the gelsolin will decrease in response due to clearance of actin–gelsolin complexes [17,18]. Plasma gelsolin levels decrease when lung tissue is damaged, and a recent study of >700 critically ill patients surprisingly found that a low plasma gelsolin level is a strong predictor of respiratory outcome, but not general outcome, in mechanically ventilated patients, and added to the value of SPD in this prediction. This must be further validated in other cohorts of critically ill ventilated patients. Non-specific markers of endothelial injury like soluble thrombomodulin and syndecan-A, do not seem to have a role in predicting acute lung injury in itself, but do predict overall prognosis in both children and adults with pre-existing respiratory failure [19,20].

## 3. Biomarkers of Lung Infection: Procalcitonin

### 3.1. Procalcitonin for Initiating Antibiotics in Critically ill Patients

Observational studies have not answered the question of whether procalcitonin (PCT)-guided antibiotic initiation and escalation could improve the survival in septic critically ill patients. This was the question posed in the Procalcitonin And Survival Study (PASS), a 1200-patient randomized controlled multi-center trial. In this trial, among the active treatment group, an increasing PCT level led to empiric broadening of antibiotics according to a specified algorithm and prompted additional culture samples and radiological imaging of suspected infected foci [21]. Although the adherence to the antimicrobial intervention algorithm was high (82%), the study intervention did not lead to a survival benefit. The principal reasons for such a result may include a “neutralizing” effect (harm from antibiotics and benefit from better antibiotic timing). Additionally, patients judged by clinicians as at high risk already received sufficient coverage of antibiotics. Thus, increasing PCT, in a broad, critically ill sepsis population, should not, by itself, lead to antimicrobial escalation. 

### 3.2. Procalcitonin for Antibiotic Reduction

PCT has been tested as a tool to discontinue antibiotics among patients with acute respiratory tract infections in settings and severities as different as primary care [22], emergency care [23,24], in patients with bacteremia [25] and intensive care [26,27]. Through all these settings, it has been proved that the duration of antibiotic treatment can be reduced substantially when the adherence to the protocolized intervention is reasonable, and when serial measurements are performed. These studies, and other alike, have been summarized in a systematic review, in which the power was sufficient to explore mortality and antibiotic side effects [28]. In this study, it was found that the defined side effects of antibiotics were reduced from 22% to 16%, and surprisingly, mortality was reduced among the PCT-guided patients. In contrast, no obvious benefit regarding antibiotic reduction was found in a recent trial in patients recruited *before* admission to hospital for acute respiratory tract infection. Procalcitonin was used to assist in the decision to admit to hospital and, for those who were admitted, to guide the length of treatment. An important limitation of the trial was that the adherence to the protocol was as low as 30%–45% among patients with low PCT (i.e., prompting antibiotic discontinuation), which may have reduced the efficacy of the strategy. 

In conclusion, PCT is an evidence-based and documented safe method of substantially reducing unnecessary use of antibiotics among hospitalized patients with acute respiratory tract infections. Such a strategy seems particularly successful, if PCT measurements are done at admission and for approximately every 24–48 h, but only if there is a relevant degree of adherence to the PCT algorithm. An example of such a discontinuation algorithm is displayed in Figure 2.

## 4. Phenotypes of Lung Inflammation and How to Use This for Improved Management

Corticosteroids are most effective in patients with eosinophilic airway inflammation, and reducing eosinophilic airway inflammation has been a target, as this has shown to reduce the risk of future exacerbations in asthma and COPD [29,30]. 

The role of eosinophils in acute inflammation, both in asthma and COPD, has been of great interest for the last decade. Recruitment of eosinophils to the airways under such conditions, is mediated by a coordinated action of cytokines and chemokines, including IL-5 and IL-13 [31]. In addition to being reflective of airway inflammation, high eosinophils in sputum and blood have been associated with frequent exacerbations, fixed airflow limitation [32] and better response to corticosteroids in patients with COPD and asthma [32,33,34]. The measurement of blood eosinophil count is much easier than collecting sputum samples, which requires more experience and time and may not always be precise and successful [35].

Approximately 20%–40% of COPD patients have eosinophilic inflammation [30], and these patients do have more frequent acute exacerbations than COPD patients without eosinophilic inflammation. COPD patients with higher blood eosinophil counts have shown to benefit more from treatment with systemic corticosteroids in moderate exacerbation of COPD compared to patients with lower blood eosinophil counts [36]. Recently, a randomized controlled trial (*n* = 318) showed non-inferiority in treating hospitalized patients with severe acute COPD exacerbation through eosinophil-guided corticosteroid treatment compared with guideline-based 5-day treatment with systemic corticosteroids, and simultaneously reducing the overall exposure to systemic corticosteroids by approximately half [37]. Also, COPD patients with lower blood eosinophil count have shown to be associated with increased risk of pneumonia, irrespective of ICS use [38], and these patients may not benefit from systemic corticosteroid treatment [36]. Therefore, this biomarker has been suggested as a tool to guide systemic corticosteroid treatment in both moderate and severe exacerbations of COPD [36,37].

## 5. Omics: Clinical Phenotypes and Advanced Bioinformatics—How to Integrate 

‘Omics’ profiling refers to the unbiased analysis of the entirety (or large percentage) of a specific class of biochemical species (e.g., DNA, RNA, lipids, metabolites or the microbiome). These analyses are becoming increasingly valuable for the development of precision medicine strategies in a variety of heterogenous diseases [39,40,41,42]. In the context of acute lung injury, omics analyses have the potential to identify novel biomarkers of acute lung damage, while also delineating specific acute lung injury endotypes that may benefit from interventional strategies targeted to the unique pathophysiology of the specific endotype. 

Currently, most omics studies in acute lung injury involve the use of animal models or cell culture, and clinically founded omics data have so far been sparse. However, it seems these lab-based models have not been particularly successful in translating into clinical benefit for acute lung injury. Therefore, they should not be used as the primary biomarker discovery platform in future omics-based research. Instead, a ‘reverse-translational’ approach could prove beneficial. Using such an approach, clinical cohorts could be used to discover novel candidate biomarkers. Such a strategy can then be further strengthened, testing the mechanistic findings in animal models and cell culture models, in which the biological functions of pathways can be further elucidated. Currently, clinical biobanks collecting plasma and bronchiolar lavage fluid from populations are being expanded, and at the same time, the costs of omics analyses are substantially decreasing. This development leads to a realistic opportunity for a much-needed shift towards human studies using omics analyses in acute lung injury research. However, as the field is still developing, this review will not discuss, in detail, all available omics-based studies or methodologies (of which there are already a number of excellent reviews [43,44,45]). Nor will it advocate for a particular technology or biochemical species, which all have their pros and cons. The decision of precisely which strategy to proceed with should depend on local expertise. Instead, this review will attempt to describe some common pitfalls in presenting and interpreting omics analyses, and we will suggest ways forward so that the field can move towards a common goal, namely the identification of novel biomarkers for acute lung injury and the delineation of specific acute lung injury endotypes.

### 5.1. Study Population—The Art of Selection

Since omics in acute lung injury is in a pioneer phase, it is even more important than in conventional biomarkers studies to have very well-defined clinical data to characterize the patients precisely. Performing advanced omics analyses without being able to link these findings to precise clinical syndromes and phenotypes seems meaningless. Thus, we strongly recommend a balanced and well-functioning collaboration between a strong team of clinically knowledgeable researchers on one side and a skilled bioinformatics team on the other side. Apart from being these capacities being present, we cannot stress enough the necessity for a respectful collaboration between them, in which both the clinical point of view and the strictly analytic point of view have a strong voice. 

The development of acute lung injury is likely due to the accumulation of a variety of endogenous and exogenous risk factors (e.g., sepsis combined with a genetic pre-disposition to develop downstream acute lung injury). By their nature, healthy controls have not been exposed to the upstream risk factors (e.g., sepsis) which are required to develop acute lung injury and it is, therefore, impossible to know whether their underlying pathophysiology would put them at greater or lesser risk of acute lung injury. Theoretically, a healthy control may well have the same downstream risk factors as the diseased patient. As such, healthy controls are not recommended, since they have not been biologically stressed to produce a certain phenotype and using healthy controls may lead to a decrease in the signal/noise ratio, if not directly to false conclusions. Instead of using healthy controls, comparisons should focus on those patients who do, in fact, develop the phenotype of interest (in this case acute lung injury) versus those with the *same exposures*, but who do not develop the acute lung injury phenotype. Such an approach was recently applied to identify variation within the SELPLG gene as a potential genetic risk factor for ARDS in African Americans [46]. Alternatively, larger studies can utilize powerful computational methodologies, such as machine learning, *latent class analysis* and *principal component analysis*, to identify biological signals that can stratify acute lung injury endotypes from a seemingly homogenous population. These studies are being increasingly applied to other heterogenous diseases and have recently shown promise in delineating sepsis endotypes [47,48,49]. While studies utilizing these methodologies in omics-based studies of acute lung injury are also beginning to appear [50,51], larger validation studies are required to confirm these early associations and confirm clinical relevance of study results.

In many clinical studies of patients with acute lung injury, all-cause mortality is used as the clinical endpoint. In omics analysis, using mortality as an outcome can lead to misinterpretation of phenotypes, since mortality can be the ultimate result of many different upstream and downstream risk factors, with many of these risk factors not being directly related to the biological processes studied (e.g., time to presentation at hospital/ICU). These risk factors are difficult to capture in clinical studies and may bias any associations discovered, thus, using mortality as a classic endpoint does not make much sense. 

### 5.2. The Importance of Validation

Regardless of the specific patient population or bioinformatics analysis chosen, the key to distilling biologically relevant signals from omics analyses is validation of novel findings in independent cohorts. In order to ensure adequate validation, omics-based studies should ensure detailed description of all methodologies employed. This should include, but is not be limited to, the ascertainment and description of clinical characteristics, the sampling procedures and time-points chosen, wet-lab processing methods, as well as any data quality control or data processing steps undertaken. For this last point, a number of useful minimum reporting guidelines for the specific omics technologies already exist and should be followed [52,53,54,55,56,57,58]. While it may seem generic, reporting of sampling relative to the event of interest is critical for an effective validation, as is reporting of potential upstream clinical confounding events that may impact downstream biology (antibiotic initiation, steroid use, fluid administration, ventilator management and vasoactive drugs). These pivotal factors are often not adequately described in study methodologies and failure to do so can lead to misinterpretation of study results. Because of these difficulties, the ideal approach for validation is multicentre collaborations or consortia where large sample sizes can be assembled and study methodologies can be standardized. An important challenge that should be addressed early in the process is the potential for data heterogeneity under multiple forms. The consortium executive committee should take responsibility for strict protocolization of all steps in sampling, aliquoting, freezing, timing of all these procedures, data extraction and entry and bio-analysis, which can often be done by using a formalized and validated platform. Multicenter clinical cohorts have been analyzed successfully in acute lung injury previously, with the identification of a hyperinflammatory ARDS sub-phenotype in two independent multicenter clinical trials [59]. Such a large-scale approach has yet to be attempted using omics technologies, but the cohorts and biological samples are available, and it is only a matter of time before similar approaches are performed using omics technologies. While large multicenter studies are ideal, this does not mean smaller studies have no place in the literature. Rather, these studies should still be reported (with appropriate methodological and clinical detail as described above), albeit with acknowledgement that any associations detected need be interpreted as exploratory. Then, these studies can act as hypothesis-generating studies, forming the basis of future larger validation works or downstream meta-analyses. 

In summary, omics-based methodologies have the potential to improve our understanding of acute lung injury endotypes, while also identifying clinically useful predictors of disease progression. However, to realize this potential, we advocate for a switch from animal models to well-characterized and appropriately controlled clinical cohorts. Additionally, we support the formation of strong multicenter clinical-bioinformatics consortia with strict protocols for all processes to increase the sample size and data validity of any omics-based analyses. We also recommend the reporting of smaller independent “acute lung injury omics” studies as exploratory, that may lead to further increasing our understanding of disease pathogenesis and improve clinical outcomes in people with acute lung injury. 

An illustration of the omics concept in acute lung injury is seen in Figure 3.

## 6. Wrap up: Biomarkers of Lung Injury

Tissue damage in the lungs can be differentiated on a cellular level by interpreting biomarkers like surfactant protein D (pneumocytes type II) and receptor for advanced glycation end-products (pneumocytes type I). The often-corresponding endothelial injury in the lung and other organs can be characterized with markers like soluble thrombomodulin and syndecan-1. Damage to the conductive airways is associated with changes in the levels of club cell secretory protein 16. 

Least but very promising, “omics” technologies may be an effective way to help us improve the understanding of the complicated disease mechanisms taking place during acute lunge injury and may offer a new way of developing new biomarkers of lung injury to assist in clinical decision-making.

Models for such tools are PCT and blood eosinophilic count: PCT has been demonstrated in several trials to help in reducing antibiotic exposure, while at the same time, reducing mortality and side effects in patients with acute respiratory tract infections. Blood eosinophilic count has just recently been demonstrated, in a safe way, to assist in reducing the exposure to systemic corticosteroids. Omics analysis is key to discovering and understanding endotypes which result in specific clinical phenotypes, and eventually, clinical outcomes. Such analyses should be performed in clinically well-characterized, preferentially large populations and be led by a strong collaboration between clinical researchers and skilled bioinformatics and founded in a consortium with a balanced participation from these two backgrounds.

## Figures and Tables

**Figure 1 jcm-08-01163-f001:**
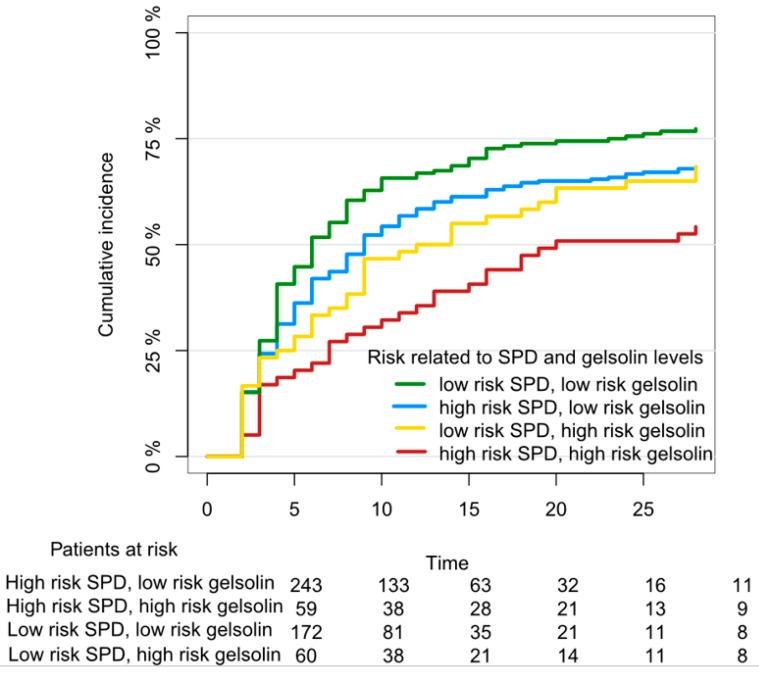
Chance for “successful weaning” from a ventilator within 28 days in critically ill patients according to blood levels of lung injury biomarkers Surfactant Protein D and Gelsolin, adapted from Holm et al. [4]. Gelsolin categories are quartiles and “high-risk gelsolin” is the *lowest* quartile, “low-risk gelsolin” was the three remaining quartiles. “High-risk SPD” was the upper 15%-percentile, “low-risk SPD” were all other SPD measurements, according to Jensen et al. [1].

**Figure 2 jcm-08-01163-f002:**
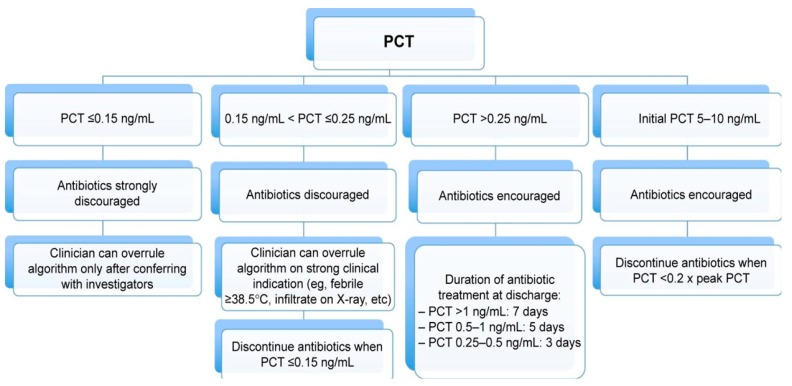
Suggestion of a PCT-guided antibiotic discontinuation algorithm based on Corti et al. [24].

**Figure 3 jcm-08-01163-f003:**
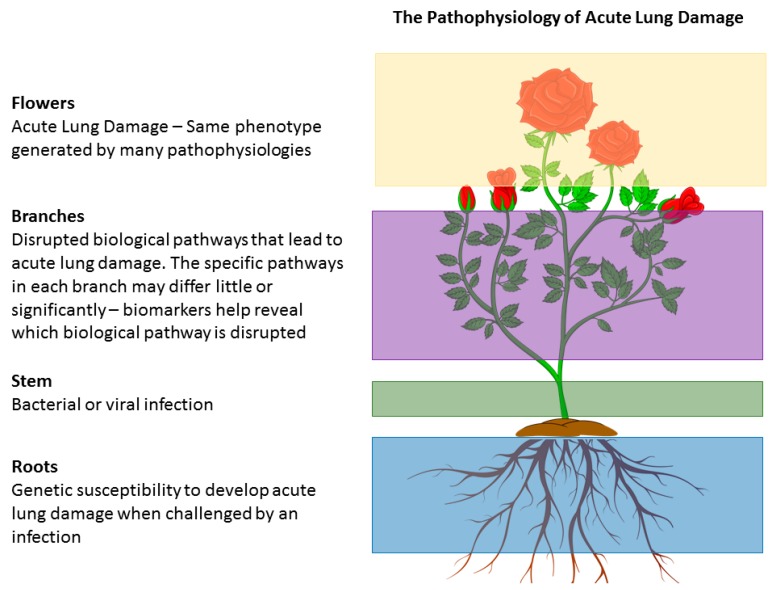
Conceptualization of ‘Omics’ approach in acute lung injury.

**Table 1 jcm-08-01163-t001:** Candidate and established biomarkers of acute lung injury.

Pathophysiological Entity for Biomarker	Biomarker	Established and Validated	Clinical Use Potential	Implemented Broadly	Included in This Review
Alveolar damage (Pneumocytes type I and II)	SPD	Yes	Risk stratification in mechanically ventilated patients	No	Yes
	s-RAGE	(yes)	?	No	Yes
	KL-6	(yes)	?	No	Yes
	FGF-7	No	No	No	No
Airway (conductive) damage	CC16	Yes	Possibly not in acute lung injury	No	Yes
Endothelial	VEGF	Yes	+	No	No
	Gelsolin	(yes) *	?	No	Yes
	sTM	(yes)	-	No	No
	Syndecan-1	No	-	No	No
Inflammation/Infection	PCT	Yes	Antibiotic reduction	Yes	Yes
	Eosinophilic granulocyte	Yes	Reduction of corticosteroid use	Yes	Yes
	IL-1β	Yes	No	No	No
	TNFα	Yes	No	No	No
	Mitochondrial DNA	No	Yes—possibly	No	No

SPD: surfactant protein D; s-RAGE: soluble receptor for advanced glycation end-products (sRAGE); KL-6: Krebs von Lungen-6; FGF-7: fibroblast growth factor-7; CC16: club cell secretory protein 16; VEGF: vascular endothelial growth factor; sTM: soluble thrombomodulin; PCT: procalcitonin; IL-1β: interleukin-1β; TNFα. * only one large study.

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
