# Peer review of "Biomarkers of Acute Lung Injury The Individualized Approach: for Phenotyping, Risk Stratification and Treatment Surveillance"

_jcm, 2019, doi:10.3390/jcm8081163_

Round 1

Reviewer 1 Report

This review seeks to survey the current state of biomarkers of acute lung injury as clinically useful diagnostic assays. 

The strengths of the paper include the clinical importance and common problem of ALI, the potential benefit of diagnostic biomarkers for clinical management, and the somewhat relevant and interesting overview provided.

There are two main weakness of the paper. The first is the structure of the presentation, which is rambling and varies from too little detail in some sections and then possibly too much in others (e.g. the PCT part). The second is the lack of scholarly depth in the review. There are numerous other candidate biomarkers that merit discussion if only to inform the reader that they failed to be satisfactory or are still under study (e.g. CRP, various cytokines, mitochondrial DNA, pro collagen, ang-2, and many others). At a minimum the authors should provide a table of current candidates and rationale. If the authors wish to focus on a promising subset and not the rest, this should be stated explicitly (as well as the reasons for the preference). This will leave the reader more informed about the field than the current version does. These weaknesses are related. It would be easier to provide a logical structure if there were more material being reviewed, as well as clear opinions offered as to what merits the readers attention and what does not.

Fig. 1 needs more detail. The legend (or text) does not explain the source of the data, how high vs low was established, how the molecules were assayed, etc. Some of the data may come from the cited reference 1 but there is no mention of gelsolin in that paper, and gelsolin is a major component of the story presented in Fig. 1.

The section on Omics is especially weak. The authors state their goal is to "

describe some common pitfalls in presenting and interpreting omics analyses

" but they only offer rather obvious generalities. Instead, specific concrete examples of the right way and the wrong way to carry out these analyses would be more informative for the reader. 

Author Response

Dear editor, dear reviewer 1,

Thank you for the comments.

Below, we have answered to the critique in a point-to-point manner.

Reviewer comments are noted C1, C2, C3 etc.

Responses are correspondingly noted R_C1, R_C2, R_C3 etc.

The paper has been revised accordingly.

We believe the paper has improved by these changes.

Reviewer 1:

C1+ C2:

(C1) This review seeks to survey the current state of biomarkers of acute lung injury as clinically useful diagnostic assays. 

The strengths of the paper include the clinical importance and common problem of ALI, the potential benefit of diagnostic biomarkers for clinical management, and the somewhat relevant and interesting overview provided.

There are two main weakness of the paper. The first is the structure of the presentation, which is rambling and varies from too little detail in some sections and then possibly too much in others (e.g. the PCT part).

(C2) The second is the lack of scholarly depth in the review. There are numerous other candidate biomarkers that merit discussion if only to inform the reader that they failed to be satisfactory or are still under study (e.g. CRP, various cytokines, mitochondrial DNA, pro collagen, ang-2, and many others). At a minimum the authors should provide a table of current candidates and rationale. If the authors wish to focus on a promising subset and not the rest, this should be stated explicitly (as well as the reasons for the preference). This will leave the reader more informed about the field than the current version does. These weaknesses are related. It would be easier to provide a logical structure if there were more material being reviewed, as well as clear opinions offered as to what merits the readers attention and what does not.

R_C1+C2:

Thank you for pointing to this. The structure has been changed to balance the part about different biomarkers to a higher extent.

The introduction has been expanded to explain the choice of the authors regarding which biomarkers to focus on.

The parts on biomarkers of infection (PCT) and inflammation (blood eosinophils) have been reduced and altered, not to be too voluminous, and to get more focused on the acute lung damage theme.

A table (table 1) of some candidate markers has been added to give the reader a better overview.

C3:

Fig. 1 needs more detail. The legend (or text) does not explain the source of the data, how high vs low was established, how the molecules were assayed, etc. Some of the data may come from the cited reference 1 but there is no mention of gelsolin in that paper, and gelsolin is a major component of the story presented in Fig. 1.

R_C3:

We apologize for this. The data come from reference 4 and 1. An explanation and the used definitions have now been added to the legend for figure 1:

Figure 1. Chance for “successful weaning” from ventilator within 28 days in critically ill patients according to blood levels of lung damage biomarkers Surfactant Protein D and Gelsolin. Adapted from Holm et al. [4]. Gelsolin categories are quartiles and “high risk gelsolin” is the lowest quartile, “low risk gelsolin” was the three remaining quartiles. “High risk SPD” was upper 15%-percentile, “low risk SPD” were all other SPD measurements, according to Jensen et al. [1]

C4:

The section on Omics is especially weak. The authors state their goal is to "describe some common pitfalls in presenting and interpreting omics analyses" but they only offer rather obvious generalities. Instead, specific concrete examples of the right way and the wrong way to carry out these analyses would be more informative for the reader. 

R_C4:

Thank you for pinpointing this. We have re-written this entire section to give more specific examples of how one could perform these analyses, as advised by the reviewer.

Please see the rewritten section in the revised manuscript.

Reviewer 2 Report

General comments:

Murray et al have summarized the current knowledge with regard to the potential biomarkers in the field of acute lung damage.

Major comments:

1) The introduction section of the manuscript is not well structured (pages 1-2). It actually starts with the question whether or not there is a need for biomarkers of lung damage and infection. Initially, the authors should define the field of interest, which is acute lung damage. What does it mean? What kind of pulmonary disorders should be considered when we speak about the acute lung damage? What kind of pathological characteristics define acute lung damage? Please revise this section and properly introduce the readers to the main topic of this review paper.

2) The large focus of the review paper was given to the “omics” approach in identification of future biomarkers. However, none of them is mentioned. Was there any potential biomarker revealed by “omics” approach in the field of acute lung damage? If so, please summarize the most promising ones.

Minor comments:

1) On the page 3 (line 60), as it was mentioned for the first time in the text, please define the ARDS abbreviation.

2) Similarly, on the page 5 (line 97), define the PCT abbreviation.

3) Page 5 (lines 110-112): there are 2 statements from the WHO, but no reference is provided. Where was this mentioned? Include the literature source.

4) The phrase “to the entire process” was written 2 times on the page 9, line 202. Please correct.

Author Response

Dear editor, dear reviewer 2,

Thank you for the comments.

Below, we have answered to the critique in a point-to-point manner.

Reviewer comments are noted C1, C2, C3 etc.

Responses are correspondingly noted R_C1, R_C2, R_C3 etc.

The paper has been revised accordingly.

We believe the paper has improved by these changes.

Reviewer 2:

C1:

General comments:

Murray et al have summarized the current knowledge with regard to the potential biomarkers in the field of acute lung damage.

Major comments:

The introduction section of the manuscript is not well structured (pages 1-2). It actually starts with the question whether or not there is a need for biomarkers of lung damage and infection. Initially, the authors should define the field of interest, which is acute lung damage. What does it mean? What kind of pulmonary disorders should be considered when we speak about the acute lung damage? What kind of pathological characteristics define acute lung damage? Please revise this section and properly introduce the readers to the main topic of this review paper.

R_C1:

Thank you for addressing this. We have now added a part to the introduction, setting the frame ofr the review. We have tried to balance it by mentioning the key features of acute lung injury and letting this part lead to the biomarker part. The introduction now reads:

Introduction; Acute Lung Injury and biomarkers to characterize this condition, and to assist in treatment strategies.

Classic acute lung injury is characterized by hypoxemia, bilateral opacities on chest imaging like chest X-ray or CT-scans and the condition should not be fully explained by cardiac failure or fluid overload. The pathophysiology is not fully understood, but it often (but not always) includes an inflammatory process in the lungs, initiated by a number of etiologies like infection (pneumonia, sepsis), aspiration (gastic juices, water etc.), trauma (lung contusion, major thoracic surgery), inhalation toxicity, blood transfusion and others. Pathologically, acute lung injury can be divided into an early phase (exudative) and a late phase (fibroproliferative), these including epithelial and endothelial damage, activation of inflammatory mediators and eventually, fibroblast activation, coagulation, cell proliferation and apoptosis pathways.

In this review, we attempt to give the reader an insight in different biomarkers of acute lung injury., and since a comprehensive review of all biomarkers of lung injury would not fit into a single article, we focus on the following areas where biomarkers may be of special importance, and in which specific biomarkers have already been thoroughly investigated: i) biomarkers of alveolar and bronchiolar damage, ii) biomarkers of endothelial damage, iii) biomarkers of lung infection, iv) biomarkers to characterize phenotypic lung inflammation. Further, we present our view of some key points in how the ‘OMICS’-era can influence this important field. We realize, our choices regarding biomarkers described could have been different. Other promising biomarkers of lung injury do, in fact, exist: we have chosen some model biomarkers, we think have a realistic clinical potential, either already established, or biomarkers that could come in clinical use within few years.”

C2:

The large focus of the review paper was given to the “omics” approach in identification of future biomarkers. However, none of them is mentioned. Was there any potential biomarker revealed by “omics” approach in the field of acute lung damage? If so, please summarize the most promising ones.

R_C2:

Upon this and other comments, the entire “omics” section has been rewritten to include also the features required in this comment.

C3:

Minor comments:

On the page 3 (line 60), as it was mentioned for the first time in the text, please define the ARDS abbreviation.

R_C3:

This has now been done. Thank you for pointing to this.

C4:

Similarly, on the page 5 (line 97), define the PCT abbreviation.

R_C4:

This has also been done now – thank you.

C5: Page 5 (lines 110-112): there are 2 statements from the WHO, but no reference is provided. Where was this mentioned? Include the literature source.

R_C5:

Upon this comment and the comment that the PCT part was too voluminous, these sentences have been deleted to focus more directly on the biomarkers and the clinical use.

C6: The phrase “to the entire process” was written 2 times on the page 9, line 202. Please correct.

R_C6:

Thank you for pointing to this. The ‘omics’ section has been entirely rewritten and this error has also been corrected.

Please see the ‘omics’ section in the revised manuscript.

Round 2

Reviewer 1 Report

The revised manuscript is substantially improved.